# Effective optimization of root selection towards improved explanation of deep classifiers

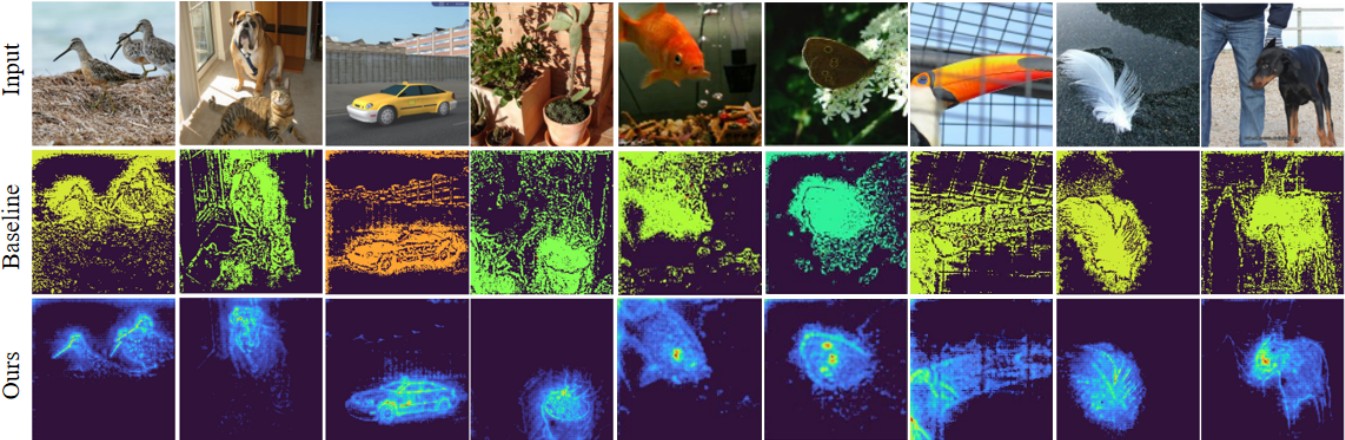

**Figure 1: Comparative illustration of heatmaps generated by our proposed and the best baseline (DTD-$z^+$) over ImageNet. As seen, the advantages achieved by our proposed can be highlighted as: (i) more accurate explanations with less noise; (ii) stronger hierarchical representation, where different regions of the target object have different colors, reflecting that the weights assigned by the proposed are closer to optimal; (iii) reflection of object contours and even textures. The visualizations utilize the Turbo colormap to highlight details and ensure color (*i.e.* relevance) accessibility for colorblind readers, and mean cropping is applied to enhance the contrast.**

## ABSTRACT

Explaining what part of the input images primarily contributed to the predicted classification results by deep models has been widely researched over the years and many effective methods have been reported in the literature, for which deep Taylor decomposition (DTD) served as the primary foundation due to its advantage in theoretical explanations brought in by Taylor expansion and approximation. Recent research, however, has shown that the root of Taylor decomposition could extend beyond local linearity, and thus causing DTD to fail in delivering expected performances. In this paper, we propose a universal root inference method to overcome the shortfall and strengthen the roles of DTD in explainability and interpretability of deep classifications. In comparison with the existing approaches, our proposed features in: (i) theoretical establishment of the relationship between ideal roots and the propagated relevances; (ii) exploitation of gradient descents in learning a universal root inference; and (iii) constrained optimization of its final root selection. Extensive experiments, including both quantitative and qualitative, validate that our proposed root inference is not only effective, but also delivers significantly improved performances in explaining a range of deep classifiers.

*ACM MM, 2024, Melbourne, Australia*

© 2024 Copyright held by the owner/author(s). Publication rights licensed to ACM.
ACM ISBN 978-x-xxxx-xxxx-x/YY/MM
https://doi.org/10.1145/nnnnnnn.nnnnnnn

## CCS CONCEPTS

• **Computing methodologies → Computer vision problems**.

## KEYWORDS

Deep Taylor decomposition, Relevance propagation, Explanation of deep classifiers

## 1 INTRODUCTION

In the realm of high-stakes decision-making, the application of Deep Learning techniques necessitates the utmost consideration for transparency, and explainability [8]. This becomes even more crucial in multimedia-centric domains, such as multimodal medical datasets analysis [40], Visual Question Answering [19] and Generative AI (Artificial Intelligence) [9]. Interpretability analysis has been widely researched over the years for visualizing those parts of input images that contributed to the predicted results of classifications [7, 11, 22, 43], where Deep Taylor Decomposition (DTD) [24] serves as one of the fundamental principles for interpretability and explainability of classifiers. In general, DTD [23, 24] is a passive interpretability method [43] that offers the advantage of explaining pre-trained neural networks without requiring any modifications to the network architecture or retraining. The essential advantage of DTD is that its explanations are mathematically proven, which

is achieved by utilizing the first-order Taylor series to approximate the propagation mapping of explanatory quantities, known as relevances between neurons. The relevance quantifies the contribution of each neuron to the overall output of the network.

However, limitations arise here due to the requirement of first-order Taylor series expansion, where the roots must be within a locally linear region to ensure its reasonable accuracy of the approximation, yet the linear region is typically small and numerous [2, 38, 42]. Further, the ideal roots should be situated at positions where the output of the network equals zero, in order to sustain that the network does not favor any specific predicted result. Although Montavon et al. [23, 24] developed layer-specific root calculation theories (referd as DTD rules) and derived the rules of Layer-wise Relevance Propagation (LRP) based on these roots, a recent study [38] has mathematically proven that these theoretically calculated roots have actually extended beyond the local linearity. Essentially, the critical issue lies in the fact that constraints on the roots are not sufficient, and thus, the selection of the rules solely relies on those relatively vague attributes without rigorous mathematical boundaries, raising the concern that human interpretation could potentially manipulate the selection process [37, 38].

In this paper, we address the above issues by extending the scope of DTD rules and mathematically establishing the relationship between roots and relevances across network layers. Correspondingly, the theoretical establishment enables us to propose an universal root calculation method for effective interpretability analysis via relevance supervised inferences. Extensive experiments validate that our proposed significantly outperforms the representative existing SoTA baselines. In summary, we highlight our contributions as follows:

- We mathematically establish the relationship between relevances and ideal roots, based on which we further propose a universal root inference scheme across layers. In comparison with the existing work, our proposed is supervised by relevances, yet the calculated roots are class-independent so that the class-related output of networks can be guided to approach zero, and the root calculation can be conducted learning with gradient descent.
- We introduce a constraint optimization of the root selection to ensure that the inferred root conforms to the requirement of Taylor expansion and the limitation of DTD.
- We carried out extensive experiments and the results validate that, in comparison with the representative existing state of the arts, the proposed achieved superior performances in both quantitative and qualitative assessments including rigorous tests of adversarial attacks, demonstrating more accurate explanations, clearer hierarchical representations, and detailed characterizations.

## 2 RELATED WORKS

Among all classification-based interpretability analysis methods that have been actively researched over the recent years, we investigate representative existing state of the arts techniques to pave the way for introducing our proposed. These include: (i) typical gradient based methods [1, 4, 17, 35, 39], such as Integrated Gradient [41], Grad-CAM [32], and SmoothGrad [27], all of which are developed out of back-propagation gradients that are initially multiplied with the input [34]. Utilizing gradients to explain individual classification decisions is a natural approach since gradients indicate the direction of fastest change on the loss function. (ii) Attribution-based methods [14, 20, 25, 26, 29] that attribute the contributions in a recursive manner from output layer to input layer. Representative methods include DeepLIFT [33], PatternAttribution [18], LIME [31], and Layer-wise Relevance Propagation (LRP) [3, 5] which involves various rules for different layers, and its main idea is to calculate the proportion (relevance) of the input multiplied by the weights in relation to the layer output.

Providing a theoretical basis for attribution-based methods [7], DTD decomposes network classification decisions (logit values of specific class) into the contributions of the elements inside input images [23]. LRP partial rules, on the other hand, are derived through the calculation of roots, and in 2022, Sixt et al. [38] have confirmed that the computation of these roots goes beyond the local linearity of first-order Taylor approximation. We have also investigated other DTD related methods such as those reported in [7, 13, 14, 16, 18, 25], but these methods adopted techniques that are well beyond DTD, making it non-comparable with our proposed.

In addition, there exist other methods that are specifically developed for certain models, such as CoDA Nets [6], DGM [36], and the recent work [7] specific for self-attention tuning. Accordingly, these model specific methods are difficult to be applied beyond their own limitations.

## 3 PROPOSED METHOD

To pave the way for our proposed research, we briefly overview the fundamental principle of DTD as follows.

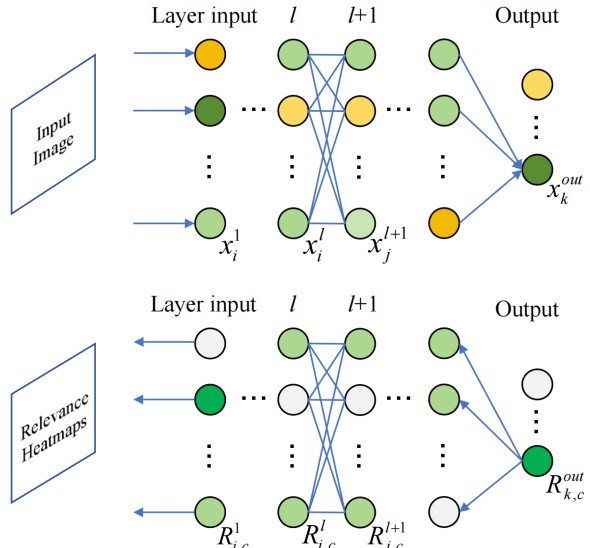

**Figure 2: Computational flow of DTD (Deep Taylor Decomposition) for explanation of deep classifications.**

In DTD, as shown in the lower part of Figure 2, the relevance for class $c$ is propagated from the output layer to the input layer via

the following formulation [24]:

$$R_{j,c}^{l+1} = \sum_i \underbrace{\frac{\partial R_{j,c}^{l+1}}{\partial x_i^l} \Big|_{\tilde{x}_i^{(j)}} (x_i^l - \tilde{x}_i^{(j)})}_{R_{i \leftarrow j,c}}, \tag{1}$$

where $\tilde{x}_i^{(j)}$ is the **root** of neuron $x_i^l$ in layer $l$ and chosen for neuron $x_j^{l+1}$ in layer $l+1$ that makes it to zero. $R_{i \leftarrow j,c}$ refers to the relevance for class $c$ that is propagated from $x_j^{l+1}$ to $x_i^l$. To this end, $R_{i,c}^l$ can be expressed as the aggregation of all $R_{i \leftarrow j,c}$ connected to $x_i^l$:

$$R_{i,c}^l = \sum_j R_{i \leftarrow j,c}. \tag{2}$$

In DTD, the propagation of relevances needs to meet:

$$\forall x : f(x) = \sum_{pixel} R_{pixel,c} \tag{3}$$

and

$$\ldots = \sum_j R_{j,c}^{l+1} = \sum_i R_{i,c}^l = \ldots = \sum_{pixel} R_{pixel,c}. \tag{4}$$

While Equation (3) ensures that "the total redistributed relevance corresponds to the extent to which the object in the input image is detected by the function" –Montavon et al. [24], Equation (4) describes the layer-wise relevance conservation law that the propagation from one layer to another is conservative in the sense of Equation (3) and $f(root) = 0$.

Inspired by the existing work that a desirable root is determined by minimizing the network's output [22, 36], and ideally, the root should make the network's output equal to zero, *i.e.* zero represents a state of ambiguity rather than positive affirmation or negative negation [22], we propose to decompose the input of a given neuron $x_i^l$ into *Signal* and *Noise*:

$$x_i^l = S_{i,c}^l + N_{i,c}^l, c \in [0, 1, \ldots, c, \ldots, M], \tag{5}$$

where subscript $c$ denotes the index of target class for which the relevance is being calculated, and $M$ represents the total number of classes. The term $S_{i,c}^l$ (*Signal*) refers an input of neuron $x_i^l$ that exclusively contains information pertaining to the target-class $c$. Conversely, $N_{i,c}^l$ (*Noise*) refers the information unrelated to that class, including the information from backgrounds and other classes. Please note that if the class shares certain characteristics with other classes (e.g. the shared feature of having two ears between dogs and cats), $S_{i,c}^l$ is then used to represent the signal after the shared characteristics being removed.

Considering the fact that, for a pre-trained robust deep classifier, noises do not affect the prediction outcome in majority cases, although they may incur interferences in the process of classification, we have:

$$f(S_{i,c}^l + N_{i,c}^l) \approx f(S_{i,c}^l) + f(N_{i,c}^l). \tag{6}$$

This can be validated by the references [15, 18], where the weights of network models are proved to be orthogonal to $N$ (noises) when training is terminated. As a result, the trained network can always maintain significant level of robustness, *i.e.* the output of class $c$ $f(S_{i,c}^l)$ dominates and $f(N_{i,c}^l)$ is close to zero, *i.e.*:

$$f(N_{i,c}^l) \approx 0. \tag{7}$$

Considering Equation (6) and inspired by the fact that the total relevance is equal to the predicted output for the target class (*i.e.* $R_{k,c}^{out} = x_k^{out}$, for the predicted class), we can establish that, for the $i$-th neuron at layer-$l$ $x_i^l$, its relevance for the target class can be derived as:

$$R_{i,c}^l = \sum_j f_{ij}^{l \to l+1}(S_{i,c}^l) \tag{8}$$

and

$$\sum_i R_{i,c}^l = \sum_i \sum_j f_{ij}^{l \to l+1}(S_{i,c}^l) = f^{l \to l+1}(S_c^l), \tag{9}$$

where $S_{i,c}^l$ represents the *Signal* for class $c$ and its associated neuron $x_i^l$ at layer $l$, and $f_{ij}^{l \to l+1}(.)$ represents the weighted function from neuron $x_i^l$ to neuron $x_j^{l+1}$.

Equation (9) can be understood as a generalization of Equation (3) for each layer. The results of $f^{l \to l+1}(S_c^l)$ accurately defines the degree (extent) to which the target object $S_c^l$ of class $c$ is detected by the layer function. In other words, it represents the sum of the neuron's relevances, following the same layer relevance conservation law as described in Equation (4).

In general, Equation (8) establishes a relationship between relevance back-propagation and forward calculation. In the forward calculation, only the *Signal* of class $c$ ($S_{i,c}^l$) is present, while the information related to other classes is close to zero. As $R_j^{l+1}$ is known, we can construct a learning process to infer the unknown variable $S_{i,c}^l$, and thus the noise $N_{i,c}^l$ can be obtained as $N_{i,c}^l = x_i^l - S_{i,c}^l$. Out of Equation (7), $N_{i,c}^l$ can be taken as the root we aim to optimize, as it removes the components from $x_i^l$ that contribute positively to the prediction of class $c$ (e.g., objects detected in an image or feature map). To minimize the deviation between the root $N_{i,c}^l$ and the original point $x_i^l$, and ensure the effectiveness of the Taylor expansion, we utilize gradient descent. This is motivated by the observation that even small perturbations in adversarial attacks can lead to changes in the predicted class. Details of its derivation are described as follows.

Denoting $T_{i,c}^l$ as the initial variable to indicate the target $S_{i,c}^l$ to be obtained, we primarily consider that the input $x_i^l$ contains the necessary information for $S_{i,c}^l$, including all patterns of interest. Correspondingly, the distance between $T_{i,c}^l$ and $S_{i,c}^l$ results in a residual $\xi$ in $R_{j,c}^{l+1}$, which can be described as:

$$R_{j,c}^{l+1} + \xi = \sum_i \underbrace{f_{ij}^{l \to l+1}(T_{i,c}^l)}_{R_{i \leftarrow j,c}}, T_{i,c}^l \Leftarrow x_i^l. \tag{10}$$

To minimize the residual $\xi$, we propose to construct the following loss and thus activate a learning process to obtain the best possible root estimation:

$$loss(T_{i,c}^l) = \xi^2 = (\sum_i f_{ij}^{l \to l+1}(T_{i,c}^l) - R_j^{l+1})^2, \tag{11}$$

where the square operation $^2$ ensures the existence of a minimum.

To minimize the loss, we apply a Taylor expansion around the poin $T_{i,c}^{'l}$ in the vicinity of $T_{i,c}^l$:

$$loss(T_{i,c}^l) = loss(T_{i,c}^{'l}) + \frac{\partial loss}{\partial T_{i,c}^l} \Big|_{T_{i,c}^{'l}} \cdot (T_{i,c}^l - T_{i,c}^{'l}) + \varepsilon \qquad (12)$$

and let

$$\triangle = T_{i,c}^l - T_{i,c}^{'l} = \gamma \cdot \frac{\partial loss}{\partial T_{i,c}^l} \Big|_{T_{i,c}^{'l}}, \gamma > 0, \qquad (13)$$

where $\triangle$ is the stride of the gradient descent, while $\gamma$ is a user-initialized parameter used to adjust the stride. Equation (13) ensures that $loss(T_{i,c}^{'l})$ is always smaller than $loss(T_{i,c}^l)$.

Correspondingly, a new $T_{i,c}^{'l}$ in which the noise is further reduced can be obtained by:

$$T_{i,c}^{'l} = T_{i,c}^l - \triangle. \qquad (14)$$

By repeating the above process iteratively, we are able to gradually bring $T_{i,c}^{'l}$ closer to the desired $S_{i,c}^l$. As a result, when the iteration concludes, $N_{i,c}^l$ can be obtained by subtracting the final $T_{i,c}^{'l}$ from $x_i^l$, as shown in Equation 5:

$$N_{i,c}^l = x_i^l - S_{i,c}^l, S_{i,c}^l \leftarrow T_{i,c}^{'l} \qquad (15)$$

To achieve the best possible explanation of the deep classification, it is crucial to determine an appropriate iteration count and stride. In practical computations, it often requires experience and multiple attempts to find the optimal values. A simple approach is to start with a smaller value and gradually increase them until the relevance heatmaps of our proposed explanation analysis no longer have any significant changes. In our experiments with the ResNet18 model on ImageNet, we set $\gamma = 1$ and perform 20 iterations, to follow the spirit of gradient based methods in adversarial attacks [21].

If it is necessary to impose a domain constraint on the computed $T_{i,c}^{'l}$, for example, we can set the following context to avoid the zero point in ReLU and ensure its differentiability:

$$T_{i,c}^{'l} \Leftarrow constraint(T_{i,c}^{'l}), \qquad (16)$$

where $\Leftarrow$ is the assignment operation, and $constraint(.)$ represents the activation function or normalization of the previous layer. By replace $T_{i,c}^{'l}$ in Equation (12) with Equation (16), we can obtain $T_{i,c}^{'l}$ within the defined domain.

Consequently, the DTD as described in Equation (2) can now be implemented in terms of our approximated and optimized root $N_{i,c}^l$:

$$R_{i,c}^l = \sum_j \frac{\partial R_{j,c}^{l+1}}{\partial x_i^l} \Big|_{N_{i,c}^l} \underbrace{(x_i^l - N_{i,c}^l)}_{R_{i \leftarrow j,c}}. \qquad (17)$$

For the convenience of implementation, finally, we summarize our proposed in Algorithm 1.

To ensure that Equation (2) can be accurately operated within a linear region of $x_i^l$, three optimizations are considered, which include: i) Initialize $S_{i,c}^l$ with zero and limit the value of $\gamma$ based on the gradient difference between $x_i^l$ and $N_{i,c}^l$ in order to prevent $N_{i,c}^l$ from going outside the linear range (short named as Gradient-constrain); ii) Consider 2nd- even high-order Taylor series; iii) Consider only the positive output of $f_{ij}^{l \to l+1}(S_{i,c}^l)$ (short named

---

**Algorithm 1** Constrained root optimization

**Input:** $x_i^l, R_{j,c}^{l+1}, step > 1, \gamma > 0$
1: Initialization: $T_{i,c}^l \leftarrow x_i^l, p \leftarrow 0, N_{i,c}^l$
2: $loss(T_{i,c}^l) = (\sum_i f_{ij}^{l \to l+1}(T_{i,c}^l) - R_{j,c}^{l+1})^2$ (via Equation (11))
3: **repeat**
4:  Calculate *gradient* of *loss* to $T_{i,c}^l$
5:  Calculate $\gamma \leftarrow$ Linear constrainted gradient clipping
6:  Calculate $\triangle \leftarrow gradient \cdot \gamma$
7:  Update $T_{i,c}^l \leftarrow T_{i,c}^l - \triangle$
8:  Update $p \leftarrow p + 1$
9: **until** $p \geq step$
10: $N_{i,c}^l \leftarrow x_i^l - S_{i,c}^l$
**Output:** $N_{i,c}^l$

---

as Positive-constrain) to avoid the nonlinear regions of neurons like ReLU and GeLU.

## 4 EXPERIMENTS AND RESULTS

In this section, we primarily focus on evaluating the explanation performances of our proposed against a number of representative existing baselines, details of which together with other essential settings of the experiments are described as follows.

**Baselines.** To address the limitations of DTD, we select the baselines based on mainstream DTD rules [24] and the LRP rules [23] that can be derived from DTD. The baselines selected include: $LRP_0$, LRP-$\gamma$, LRP-$\epsilon$, DTD-$\omega^2$, DTD-$z^+$, and DTD-$z^\beta$, among which the DTD-$z^+$ rule is equivalent to LRP-$\alpha\beta$ where $\alpha = 1, \beta = 0$. Besides, Montavon et al. [23] systematically unified the theories of DTD and LRP, and proposed a combined usage guideline (short named as Combination) which is also included. In addition, we also compared with the latest SoTA methods, including: interpretable Vision-Transformer (short named as I.ViT) [7], AGF (Attribution Guided Factorization) [14], DGM ( Deep Geometric Moment) [36], and interpretable CoDA-Net [6]. Among them, I.Vit and AGF use DTDs as components or bases. Our baseline selection is an integrated outcome of considering all aspects of the existing work and maintaining an appropriate balance to sustain a comprehensive coverage of DTD and its variations.

**Datasets.** ImageNet (ILSVRC2012, consisting of 1000 classes) is used for quantitative evaluation, pixel masking evaluation, and adversarial attack evaluation guided by heatmaps. The PASCAL-VOC 2012 (VOC'12, 20 classes) [10] and ImageNet-Segmentation-300 (ImageNet-S300, 300 classes) [12] are used for segmentation evaluation.

**Models.** To verify the generalizability of our method, we conducted experiments on different models, including: ResNet, VGG, and ViT, which are widely utilized in the field of Deep Learning, especially deep classifications.

To improve public accessibility, the source code will be made available after the paper is accepted for publication.

### 4.1 Improvement relative to original DTDs

**Positive and negative perturbation.** The positive perturbation experiment involves sequentially masking (setting to zero) pixel


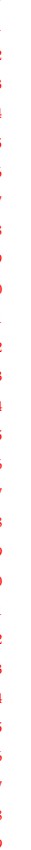
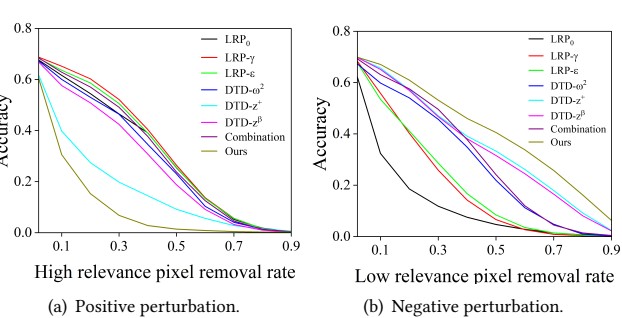

(a) Positive perturbation.

(b) Negative perturbation.

**Figure 3: Experimental results in accuracy degradation curves of positive (a) and negative (b) perturbations.**

values based on their relevance rankings, starting from the highest relevance and proceeding to the lower relevance. The accuracy drop of the model serves as a validation of the contribution made by the masked pixels, allowing for assessing whether or not the key pixels are highlighted by the interpretability methods [7]. The experimental results are presented in Figure 3(a).

In contrast, negative perturbation experiments are conducted to verify that the pixels not highlighted by heatmaps are indeed irrelevant to the concerned class. Therefore, in negative perturbation experiments, it is desirable for the model's accuracy to decrease at a slower rate as the pixels are masked in an ascending order of their relevances, indicating that those pixels are indeed unrelated to the class [7, 14]. The corresponding results are presented in Figure 3(b).

The perturbation is divided into 10 gradients, ranging from masking 1%, 10%, 20% to 90% of the pixels. Classification tests are conducted for every gradient on the ImageNet test dataset and the average accuracy is recorded. It shoud be noted that our pre-trained model, *i.e.* ResNet18, achieves an original average accuracy of 0.71 on the ImageNet test set.

As seen from Figure 3, our proposed significantly outperforms the existing baselines on positive perturbation experiments, which validate that our proposed is indeed capable of capturing the key pixels relevant to the concerned class. For the negative perturbation experiment, our proposed also outperforms the selected baselines, although with less significance. This suggests that our proposed, like those selected baselines, may have missed some class-relevant pixels or regions. Compared with the existing SoTA relus, nonetheless, our proposed still remains the optimal choice. Among the numerous LRP rules derived from DTD, the DTD-$z^+$ (LRP-$\alpha\beta$) rule exhibits suboptimal results. In addition, the two perturbation experiments further indicate that the LRP$_0$ rule shows the poorest overall performance, especially in terms of the fastest accuracy decline during the negative perturbation experiment.

**Semantic segmentation.** In this section, we focus on quantitatively evaluating the quality of generated heatmaps using metrics and datasets specific to the field of semantic segmentation. It involves comparing the relevance heatmaps with the ground truth to examine whether or not the heatmaps accurately cover class-relevant objects.

**Table 1: Semantic segmentation metrics in percentage on ImageNet-S300 and VOC'12 datasets.**

|  | LRP$_0$ | LRP-$\gamma$ | LRP-$\epsilon$ | DTD-$\omega^2$ | DTD-$z^+$ | DTD-$z^\beta$ | Combination [23] | Ours |
|---|---|---|---|---|---|---|---|---|
| | | | | ImageNet-S300 | | | | |
| mPA | 50.22 | 49.83 | 48.39 | 50.94 | 52.30 | 50.04 | 48.30 | **68.39** |
| mIoU | 45.93 | 46.49 | 46.58 | 47.67 | 60.83 | 46.68 | 46.26 | **69.44** |
| | | | | VOC'12 | | | | |
| mPA | 48.46 | 49.54 | 47.62 | 49.45 | 60.15 | 50.42 | 49.62 | **74.12** |
| mIoU | 34.40 | 31.87 | 30.12 | 39.47 | 52.71 | 33.93 | 32.43 | **66.56** |

To ensure that relevance heatmaps exhibit prominent semantic contours, we process all the relevance matrices (heatmaps) in the following steps [7, 14]: (i) the relevance matrices are subjected to min-max normalization; (ii) the elements in the relevance matrix that are below the mean value are set to zero to diminish weak correlations that do not exhibit visually perceptible effects; (iii) the resulting relevance matrices are compared against the ground truth segmentation from VOC'12 [10] and ImageNet-S300 [12], enabling the computation of mIoU (mean Intersection over Union) and mPA (mean Pixel Accuracy) scores.

For Imagenet-S300, which consists of identical image copies as ImageNet, we utilize the pre-trained ResNet18 model to generate class-specific interpretable heatmaps for all assessed models. For VOC'12, considering that a majority of the classes are already included in ImageNet (except for Cow, Horse, and ambiguous), we did not train a separate model on this dataset. Instead, we directly employ the pre-trained ResNet18 to assess the cross-dataset capabilities for both the assessed models and our proposed interpretability methods. We use the training sets of VOC'12 to generate heatmaps and perform segmentation evaluation.

The results of the semantic segmentation evaluation are presented in Table 1, which evidence that our proposed achieves the best performances in terms of segmentation over both datasets. In terms of mPA, which measures the proportion of correctly highlighted class-relevant pixels with respect to the total number of labeled pixels in the dataset, our proposed exhibits higher pixel-level accuracy compared to the baselines, highlighting its advantage in capturing fine-grained details. In terms of mIoU, which provides a measure of spatial overlap between the heatmap and the ground truth, our proposed improves the ability to capture the shape and boundaries of the target objects in comparison with those existing baselines.

In addition, the experiments also show that the DTD-$z^+$ rule (LRP-$\alpha\beta$) has achieved suboptimal performances, which is consistent with the observation in the positive and negative perturbation experiments.

**Relevance guided adversarial attacks.** Inspired by the gradient-based attack as reported in [21, 28], we carry out further experiments by employing interpretability heatmaps to guide pixel selection in non-target adversarial attacks. This allows us not only to quantitatively evaluate the quality of generated heatmaps based on the Attack Success Rate (ASR) scores, but also enables exploration of the application values of the interpretability research in adversarial attacks and defenses.

**Table 2: Results of the adversarial attack experiment on ImageNet. The term "Step" refers to the average iterations required for a successful attack, and "Acc" refers the post-attack accuracy. A lower value is preferred for both "Step" and "Acc".**

|  | $LRP_0$ | LRP-$\gamma$ | LRP-$\epsilon$ | DTD-$\omega^2$ | DTD-$z^+$ | DTD-$z^\beta$ | Combination [23] | Ours |
|---|---|---|---|---|---|---|---|---|
| ASR | 0.40 | 0.24 | 0.31 | 0.37 | 0.40 | 0.40 | 0.37 | **0.57** |
| Step | 14.54 | 13.91 | 14.15 | 15.33 | 12.90 | 14.62 | 13.87 | **7.24** |
| Acc | 0.42 | 0.54 | 0.49 | 0.45 | 0.43 | 0.43 | 0.45 | **0.31** |

Our experiment is based on the traditional PGD (Projected Gradient Descent) [21] and incorporates techniques similar to JSMA (Jacobian-based Saliency Map Attack) [28] to reduce the perturbation search space of PGD from the entire image to a reduced area of 0.003 (500 pixels in RGB space $224 \times 224 \times 3$) in accordance with the corresponding heatmaps.

The experiment consists of three stages, including: (i) computing the interpretability heatmaps for the pre-trained ResNet18 model on the ImageNet test set; (ii) applying PGD [21] to the 500 pixels with high rankings in terms of relevances in the heatmaps and generate adversarial samples; (iii) computing the model's re-predictions on the adversarial samples. This is an iterative process that continues until either the model makes an incorrect prediction or reaches the iteration limit, which is set as 20 in this work. The corresponding results are summarized in Table 2.

To minimize the perceptibility of adversarial samples, we employ the $l_\infty$ norm to constrain the range of pixel perturbations. Specifically, we set the maximum perturbation level for a single pixel to be 0.03, which corresponds to 8/255. Additionally, the use of heatmaps introduces an $l_0$ norm constraint, which limits the number of perturbed pixels.

The results in Table 2 demonstrate that our heatmap provides more accurate guidance for PGD attacks, resulting in the highest ASR score 0.57 achieved in the experiment, which means more than half of the samples were misclassified by the model due to a minor perturbations (smaller than 0.03 in magnitude) on only 0.003 of the total pixels. In contrast, the highest ASR score for other baselines does not exceed 0.4, and the average iterations per successful attack is also twice times more than our proposed, indicating that the pixels provided to PGD have low correlations with the target class, resulting in attack failure.

As seen, the score for the LRP-$\gamma$ rule is the lowest, which aligns with the results observed in the positive perturbation experiment (Figure 3(a)).

## 4.2 Qualitative evaluation

To facilitate the visualization of relevance, we employ Turbo colormap [30] which provides a smooth transition of colors and highlights details effectively to facilitate our qualitative evaluations. As Turbo colormap is primarily designed to be colorblind friendly, we further apply mean cropping [14] to enhance the contrast for visualizations. This involves removing the portions of correlated heatmaps that have values below the mean.

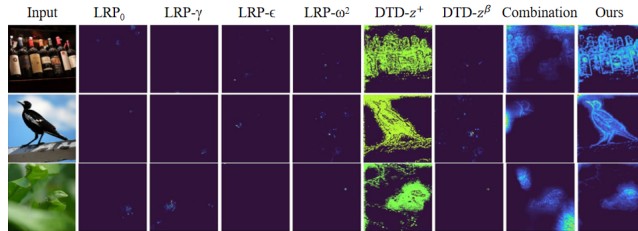

**Figure 4: Illustration of heatmap comparison between our method and the baselines.**

The visualization results on ImageNet dataset are illustrated in Figure 1 and Figure 4. Consistent with the quantitative experiments, as seen, the DTD-$z^+$ rule achieved suboptimal performances. DTD-$z^+$ and Combination [23] exhibited recognizable patterns. The heatmaps generated by other rules, however, showed almost no visually meaningful patterns, and their heatmaps align with the visualization results conducted in a recent work [7]. Figure 1 highlights more comparisons with the suboptimal method DTD-$z^+$, and it superiority over other rules, as attributed by Sixt et al., is related to the working principles of convolution, further details of which are referred to their recent work [38].

It is evident that our method is superior to all baselines as shown in Figure 1, which demonstrates three distinct advantages of our approach. Additionally, Figure 4 showcases this kind of advantages in three typical scenarios: clearer and more comprehensive explanations in multi-object scenes (first row), outstanding ability to characterize details of the target (second row), and explanations even in complex backgrounds that are difficult for the human eyes to discern (third row).

As a matter of fact, our multiple experiments support that the interpretability advantage of our proposed, which is showcased in Figure 1 and Figure 4, is not an isolated case, but rather a consistent pattern that can be observed across the majority of samples for all categories. More samples are provided in the supplementary material for further examinations, and our experimental codes are also made publicly available for further references and verifications.
**Layer-wise Interpretability.** In principle, the layer-by-layer propagation of relevances in DTD can be exploited to enable a layer-wise interpretation of the concerned neural networks. To the best of our knowledge, however, we have not found any attempt of such work in our latest literature survey.

To facilitate a layer-wise interpretability analysis, we firstly extract relevances from the 2nd, 4th, 6th, 8th, 10th, and 12th layers of ResNet18 with equal spacings, respectively. After summing the values along the channel and then subject them to normalization, we apply a process of Turbo color mapping to produce the visualization results, as shown in Figure 5. Note that no mean cropping is applied here, not even for heatmaps.

Out of the visualization results in Figure 5, an interesting analysis can be conducted. In the first row, as seen, the pixel-level interpretation of the airplane (*i.e.* heatmaps) exhibits some noises and roughly depicts the boundary of the clouds in the upper right corner. Through comparative examination across layers, however, conclusive observations can be made that it is not the boundary

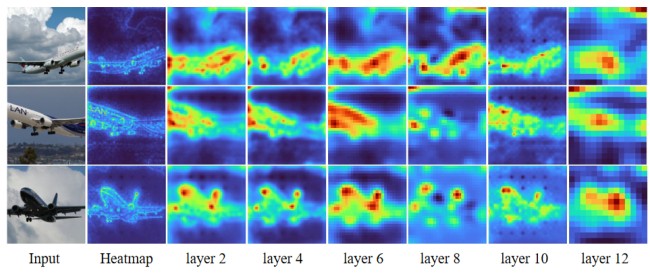

**Figure 5: Visualization of the precise layer-wise interpretability of airplane images indicate that ResNet18 has learned the coupling relationship between airplanes and the sky. By examining the interpretation results at different layers, we can gain insights into how the network perceives and represents this coupling.**

of the clouds but the boundary of the sky. We speculate that this might be because the neural network has observed a frequent co-occurrence of the sky and airplanes, making the sky a useful context for predicting airplanes. To support this hypothesis, we further investigated all the explanation heatmaps for this class and confirmed that this is true. In addition, it can be seen that the upper edges of the airplane often exhibit significant correlations. This can also be established for all other visualizations as shown in the remaining rows of Figure 5.

### 4.3 Whether $f(N)$ close to 0?

For the $S$ trained hence robust network, noises may introduce interference in its output but not able to affect its prediction outcome, *i.e.* for a network $f$ taking $S$ as input matrix, $f(S + N)$ can be approximated by $f(S) + f(N)$ in Equation (6). A small value of $f(N)$ close to 0 guarantees that the residuals of the Taylor expansion are negligible, thus preserving the layer-wise relevance conservation in Equation (4). Consequently, we evaluated the output of the target class when the network $f$ took the raw image $X$, signal $S$, and noise $N$ as the input. The results are illustrated in Figure 6, where the output $f(N)$ consistently exhibiting a small value near 0.

We further demonstrated the visualization results of roots and signals from the hidden layers, as shown in Figure 7, we can intuitively observe the visual representations of the root and *Signal* and assess if they align with our expectations or not. Note that a 0-255 normalization and Gaussian smoothing have been applied for all feature maps, and the colors do not represent their actual numerical values. As seen from Figure 7, *Signal* exhibits a clear visual structure, depicting the main outline of the object of interest. In contrast, visualization of roots as shown in Figure 7 can be seen as highlighting the pixel regions outside the object of interest, which can be called as class-independent *Noise*. These qualitative results validate the proposed schemes as in section 3. Combined with the results of quantitative assessment achieved from the experiments in section 4.2, it can be established that our proposed interpretability method indeed operates in accordance with the theoretical design.

It is worth noting that the visualization of *Signal* in Figure 7 still contains some noises, which may arise from the failure of loss function and its optimization as described in section 3.3 in obtaining

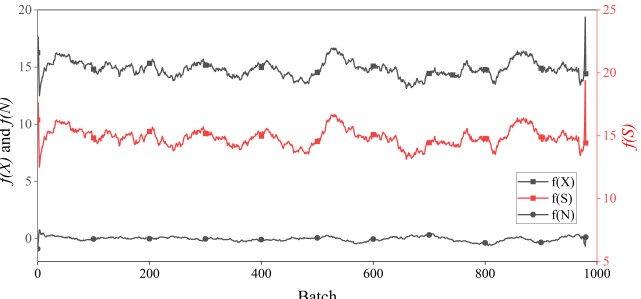

**Figure 6: Output Values of the target class for different inputs: original image $X$, Noise $N$ (vertical axis on the left), and Signal $S$ (vertical axis on the right). These outputs are obtained from ResNet18 applied to the ImageNet validation set. To enhance clarity, all curves have been smoothed using a window size of 20.**

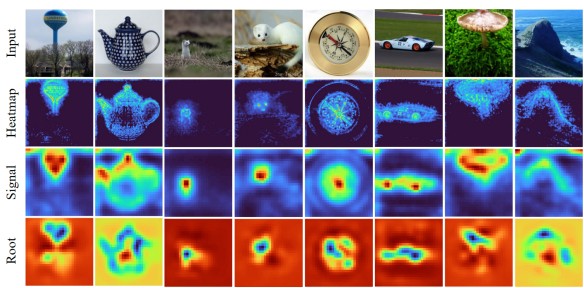

**Figure 7: Visualization of signals and roots in the 8-th convolutional layer of ResNet18 for objects of different categories, where heatmaps of *Signal* and roots are generated with Gaussian smoothing.**

the optimal *Signal*. Other possibilities include the residual in Taylor approximations, or even the noise brought in by network weights. As a result, there still exist some spaces for further research upon our proposed, including optimizing loss functions etc., and we will address these limitations in our future work.

### 4.4 Compare to SoTAs on different models

To verify that our method can also improve the performance of existing DTD based methods, and to demonstrate the generalizability of our method on different models. We evaluated the heatmaps on two other models: VGG-19, and ViT. The baseline method of VGG-19 involves AGF [14], which is an interpretable method based on DTD, so we can use our method to improve the calculation of the DTD part. Similarly, the interpretation of ViT utilized the SoTA method (I-ViT) [7] based on DTD, which is a interpretable approach specifically optimized for the Transformer architecture. We also used ImageNet-S and VOC datasets here. The relevant results are shown in Table 3 and Table 4.

In order to provide a comprehensive coverage of the latest methods in the field, we also compared our approach with the SoTA works from the past two years, regardless of whether they were based on a DTD or not. Our investigation included: DGM [36] (based

**Table 3: Comparison with several existing DTD-based ViT interpretation methods.**

|  | Metrics | rollout | raw attention | I.ViT [7] | Ours |
|---|---|---|---|---|---|
| ImageNet-S300 | mPA | 65.15 | 67.84 | 70.59 | **73.26** |
|  | mIoU | 55.42 | 46.37 | 81.97 | **82.51** |
| VOC'12 | mPA | 62.76 | 64.97 | 69.25 | **71.83** |
|  | mIoU | 49.85 | 38.46 | 73.09 | **75.14** |

**Table 4: Comparison with several existing DTD-based CNN(VGG-19) interpretation methods.**

|  | Metrics | GradCAM | SmoothGrad | AGF [14] | Ours |
|---|---|---|---|---|---|
| ImageNet-S300 | mPA | 59.58 | 62.27 | **71.41** | 67.37 |
|  | mIoU | 57.36 | 54.41 | 64.25 | **69.88** |
| VOC'12 | mPA | 61.85 | 56.92 | 63.16 | **66.52** |
|  | mIoU | 38.04 | 30.39 | **68.54** | 65.95 |

**Table 5: Comparison with two latest XAI.**

|  | Metrics | CoDA-Net [6] | DGM [36] | Ours |
|---|---|---|---|---|
| ImageNet-S300 | mPA | 62.26 | 65.41 | **68.39** |
|  | mIoU | 58.18 | 69.10 | **69.44** |
| VOC'12 | mPA | 64.25 | 70.41 | **74.12** |
|  | mIoU | 64.88 | **73.53** | 66.56 |

on ResNet18) and CoDA-Net [6]. The two works are examples of explainable artificial intelligence (XAI) that incorporate specifically designed network architectures with self-explanatory capabilities. We applied our method to interpret the classification of ResNet18 and the comparison with their method can be found in Table 5.

## 4.5 Ablation study

To assess the effectiveness of the three measures proposed in Section 3, namely Gradient-constrain, Positive-constrain, and 2nd-Taylor, in avoiding the root $N_{i,c}^l$ from exceeding the linear neighborhood of $x_i^l$, we conducted ablation experiments for each of these measures. The aim was to determine whether these measures contribute to improved explanations.

Generally, verifying whether $x_i^l$ and $N_{i,c}^l$ are within the linear region requires confirming the equality of their gradients and the existence of a gradient equality path between them. However, for the sake of simplicity, in the case of the GeLu activation used in ViT, we only verify the gradient equality.

The ablation experiments were conducted in three steps: (i) The relevant measures were individually masked; (ii) The segmentation evaluation for relevance heatmaps were performed on VOC'12 and ImageNet-S300 using ResNet18; (iii) The batch-averaged Mean-Squared Error (MSE) of the gradients of $N_{i,c}^l$ and $x_i^l$ for all neurons in the frist fully-connected layer with GeLu activation in ViT was evaluated. The results are presented in Table 6.

Table 6 have revealed that the gradient difference between the roots $N_{i,c}^l$ and $x_i^l$ is very small for most neurons (Gradient-MSE is a average of batch), indicating that they are likely situated within a

**Table 6: Results of Ablation study. Gradient-MSE indicates the mean-square error of gradients between $x_i^l$ and the root $N_{i,c}^l$. In general, a smaller Gradient-MSE indicates that the root is closer to $x_i^l$, resulting in better DTD. Conversely, a larger MSE value after ablation indicates the importance of the specific module. The symbol / indicates that the ablation of this item does not affect the result.**

|  | Metrics | Posotive -constrain | 2nd-Taylor | Gradient -constrain | Before ablation |
|---|---|---|---|---|---|
| ImageNet-S300 | Gradient-MSE | 0.081 | / | 0.233 | 0.071 |
|  | mPA | 63.97 | 68.13 | 66.23 | 68.39 |
|  | mIoU | 58.05 | 69.29 | 62.60 | 69.44 |
| VOC'12 | Gradient-MSE | 0.099 | / | 0.242 | 0.095 |
|  | mPA | 69.85 | 74.04 | 72.55 | 74.12 |
|  | mIoU | 60.77 | 65.39 | 61.48 | 66.56 |

linear region. Table 6 also shows that considering only positive values in DTD is highly efficient, which could explain the suboptimal performance of the DTD-$z^+$ rule in Section 4.1.

## 5 CONCLUSTIONS

Over recent years, DTD has been widely researched to ascertain its theoretical advantages in providing explanations for neural network based classifications via exploitation of Taylor polynomial approximations. Recent research, however, has raised concerns about its interpretability and explainability, primarily due to its limitations of local linearity and its lack of theories to ensure reliable selection of ideal roots.

In this paper, we address those limitations of DTD by introducing: a universal root inference and calculation scheme via relevance supervised learning over gradient descent. In comparison with the existing approaches, our proposed achieves the advantage that the constraint optimization allows a class-independent root selection, and the proposed gradient-based learning makes it optimally approach its ideal estimation, thereby delivering its expected interpretability performances.

Extensive quantitative experiments and qualitative analysis, including evaluations in semantic segmentation and heatmap guided adversarial attacks, validate that the proposed outperforms the existing baselines in providing more precise, robust, and comprehensive explanations. Future work can be directed to focus on complexity optimization to further improve the performances on interpretability analysis.

In summary, our contributions described in this paper provide a successful solution to the shortfall and limitations of DTD. It not only improves its practical interpretability performances but also provides DTD with an essential foundation for its further application and expansion in developing universal interpretable methods for all continuously differentiable deep networks.

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
