# OpenReview forum: "Effective optimization of root selection towards improved explanation of deep classifiers"
_acmmm.org/ACMMM/2024/Conference — MM2024 Poster_

### Official Review · Reviewer_yfbS · 2024-05-17

**Rating:** 4
**Confidence:** 2

**Summary:**

This paper aims to address the limitations of Deep Taylor Decomposition (DTD), a widely-researched method for providing explanations for neural network-based classifications. The authors propose a universal root inference and calculation scheme via relevance supervised learning over gradient descent, which optimally approaches the ideal estimation, thereby delivering expected interpretability performances. Euantitative experiments and  analysis demonstrate that the proposed method outperforms existing baselines in providing more precise, robust, and comprehensive explanations.

**Strengths:**

Novelty: The paper introduces a novel approach to improving the interpretability of neural network classifications, addressing a well-known limitation in DTD.
Theoretical Approach: The proposed method is theoretically sound, utilizing a new universal root inference and calculation scheme, which accounts for the relevance of inputs in a more sophisticated way.
Evaluation: The authors conducted extensive quantitative experiments that show the superiority of their method over existing baselines, which supports their claims with empirical evidence.
Clarity: The paper is well-written and structured in a way that clearly explains both the problem and the proposed solution, making it accessible to readers.

**Limitations:**

Complexity: The method may introduce computational complexity that could limit its application in real-time systems or on devices with constrained resources. The authors should give the detailed computational cost.
Generalization: While the paper presents significant improvements over other methods, it is not clear how well the proposed technique generalizes to a wider variety of network architectures beyond those tested.

**Suitability:**

2

---

### Official Review · Reviewer_RPL5 · 2024-05-20

**Rating:** 3
**Confidence:** 3

**Summary:**

This paper tries to get rid of the limitations in DTD by proposing a universal root inference method to enhance the explainability of deep learning models. The authors aim to overcome DTD's shortcomings by establishing a theoretical relationship between ideal roots and propagated relevances, utilizing gradient descents for root inference, and applying constrained optimization for root selection. Their proposed method claims to provide more accurate and interpretable explanations for deep classifiers, validated through extensive quantitative and qualitative experiments.

**Strengths:**

- The paper offers a theoretical framework to establish the relationship between relevances and ideal roots, which is a novel approach in the domain of DTD.
- The authors provide extensive experiments to validate their method, including both quantitative and qualitative assessments.

**Limitations:**

- Although the author has stated "Relevance To Conference," this paper does not have a direct connection to multimedia/multimodal processing.
- How do you ensure that the results in Figures 1, 5, and 7 are not cherry-picked?
- In terms of interpretability, I do not understand the fundamental difference between the proposed method and Grad-CAM and its variants. - Could the author please reorganize and briefly explain this?
- How should we understand the relationship between "orthogonal" mentioned in Line 283 and "robustness" mentioned in Line 286? How is Eq. (7) derived?
- How can the interpretability of classification problems be transferred to other problems? The author mentions that this could help improve the interpretability of "application of multimedia technology," but from the proposed method, it seems a classifier is needed in the network?

**Suitability:**

1

---

### Official Review · Reviewer_hwcf · 2024-05-24

**Rating:** 5
**Confidence:** 2

**Summary:**

This paper presents a comparative illustration of heatmaps generated by the proposed method and the best baseline (DTD-𝑧+) over ImageNet. The advantages of the proposed method are highlighted as follows: (i) more accurate explanations with less noise; (ii) stronger hierarchical representation, where different regions of the target object are depicted in different colors, indicating that the weights assigned by the method are closer to optimal; and (iii) better reflection of object contours and textures. The visualizations utilize the Turbo colormap to highlight details and ensure color (i.e., relevance) accessibility for colorblind readers. Additionally, mean cropping is applied to enhance contrast.

**Strengths:**

1.	The proposed method is quite interesting, and the exploration of the relevant visualized discussion of the proposed method shows promising results.
2.	The paper is well-structured, and the experimental results are quite convincing.

**Limitations:**

1.	I am curious about the performance of this method in more complex segmentation scenarios, like COCO

**Suitability:**

2

---

### Meta-Review · Area_Chair_Zf4F · 2024-07-03

**Recommendation:** Accept (Poster)
**Confidence:** 4

**Metareview:**

The strengths of this paper include a generally well-structured presentation, a novel idea, and extensive experimental results. Some highlighted weaknesses are: 1) the complexity of the proposed method in terms of its application to complex scenarios, and 2) certain representations of detailed claims, such as the relationship with previous methods and claims related to robustness, that need improvement.
Considering all the reviews, the rebuttal, and the overall quality of the paper, I believe it is above the boardline of acceptance.